# Gaussian process based nonlinear latent structure discovery in multivariate spike train data

**Anqi Wu, Nicholas A. Roy, Stephen Keeley, & Jonathan W. Pillow**
Princeton Neuroscience Institute
Princeton University

## Abstract

A large body of recent work focuses on methods for extracting low-dimensional latent structure from multi-neuron spike train data. Most such methods employ either linear latent dynamics or linear mappings from latent space to log spike rates. Here we propose a doubly nonlinear latent variable model that can identify low-dimensional structure underlying apparently high-dimensional spike train data. We introduce the *Poisson Gaussian-Process Latent Variable Model* (P-GPLVM), which consists of Poisson spiking observations and two underlying Gaussian processes—one governing a temporal latent variable and another governing a set of nonlinear tuning curves. The use of nonlinear tuning curves enables discovery of low-dimensional latent structure even when spike responses exhibit high linear dimensionality (e.g., as found in hippocampal place cell codes). To learn the model from data, we introduce the *decoupled Laplace approximation*, a fast approximate inference method that allows us to efficiently optimize the latent path while marginalizing over tuning curves. We show that this method outperforms previous Laplace-approximation-based inference methods in both the speed of convergence and accuracy. We apply the model to spike trains recorded from hippocampal place cells and show that it compares favorably to a variety of previous methods for latent structure discovery, including variational auto-encoder (VAE) based methods that parametrize the nonlinear mapping from latent space to spike rates with a deep neural network.

## 1 Introduction

Recent advances in multi-electrode array recording techniques have made it possible to measure the simultaneous spiking activity of increasingly large neural populations. These datasets have highlighted the need for robust statistical methods for identifying the latent structure underlying high-dimensional spike train data, so as to provide insight into the dynamics governing large-scale activity patterns and the computations they perform [1–4].

Recent work has focused on the development of sophisticated model-based methods that seek to extract a shared, low-dimensional latent process underlying population spiking activity. These methods can be roughly categorized on the basis of two basic modeling choices: (1) the dynamics of the underlying latent variable; and (2) the mapping from latent variable to neural responses. For choice of dynamics, one popular approach assumes the latent variable is governed by a linear dynamical system [5–11], while a second assumes that it evolves according to a Gaussian process, relaxing the linearity assumption and imposing only smoothness in the evolution of the latent state [1, 12–14]. For choice of mapping function, most previous methods have assumed a fixed linear or log-linear relationship between the latent variable and the mean response level [1, 5–8, 11, 12]. These methods seek to find a linear embedding of population spiking activity, akin to PCA or factor analysis. In many cases, however, the relationship between neural activity and the quantity it encodes can be highly nonlinear. Hippocampal place cells provide an illustrative example: if each discrete location in a 2D

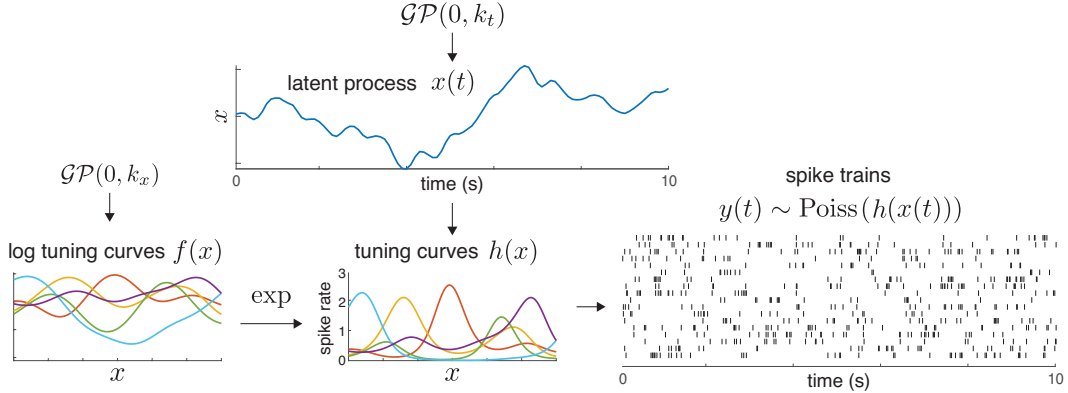

Figure 1: Schematic diagram of the Poisson Gaussian Process Latent Variable Model (P-GPLVM), illustrating multi-neuron spike train data generated by the model with a one-dimensional latent process.

environment has a single active place cell, population activity spans a space whose dimensionality is equal to the number of neurons; a linear latent variable model cannot find a reduced-dimensional representation of population activity, despite the fact that the underlying latent variable ("position") is clearly two-dimensional.

Several recent studies have introduced nonlinear coupling between latent dynamics and firing rate [7, 9, 10, 15]. These models use deep neural networks to parametrize the nonlinear mapping from latent space to spike rates, but often require repeated trials or long training sets. Table 1 summarizes these different model structures for latent neural trajectory estimation (including the original Gaussian process latent variable model (GPLVM) [16], which assumes Gaussian observations and does not produce spikes).

| model | latent | mapping function | output nonlinearity | observation |
|---|---|---|---|---|
| PLDS [8] | LDS | linear | exp | Poisson |
| PfLDS [9, 10] | LDS | neural net | exp | Poisson |
| LFADS [15] | RNN | neural net | exp | Poisson |
| GPFA [1] | GP | linear | identity | Gaussian |
| P-GPFA [13, 14] | GP | linear | exp | Poisson |
| GPLVM [16] | GP | GP | identity | Gaussian |
| **P-GPLVM** | **GP** | **GP** | **exp** | **Poisson** |

Table 1: Modeling assumptions of various latent variable models for spike trains.

In this paper, we propose the *Poisson Gaussian process latent variable model* (P-GPLVM) for spike train data, which allows for nonlinearity in both the latent state dynamics and in the mapping from the latent states to the spike rates. Our model posits a low-dimensional latent variable that evolves in time according to a Gaussian process prior; this latent variable governs firing rates via a set of non-parametric tuning curves, parametrized as exponentiated samples from a second Gaussian process, from which spikes are then generated by a Poisson process (Fig. 1).

The paper is organized as follows: Section 2 introduces the P-GPLVM; Section 3 describes the *decoupled Laplace approximation* for performing efficient inference for the latent variable and tuning curves; Section 4 describes tuning curve estimation; Section 5 compares P-GPLVM to other models using simulated data and hippocampal place-cell recordings, demonstrating the accuracy and interpretability of P-GPLVM relative to other methods.

## 2 Poisson-Gaussian process latent variable model (P-GPLVM)

Suppose we have simultaneously recorded spike trains from $N$ neurons. Let $\mathbf{Y} \in \mathbb{R}^{N \times T}$ denote the matrix of spike count data, with neurons indexed by $i \in (1, \ldots, N)$ and spikes counted in discrete

time bins indexed by $t \in (1, \ldots, T)$. Our goal is to construct a generative model of the latent structure underlying these data, which will here take the form of a $P$-dimensional latent variable $\mathbf{x}(t)$ and a set of mapping functions or tuning curves $\{h_i(\mathbf{x})\}$, $i \in (1, \ldots, N)$ which map the latent variable to the spike rates of each neuron.

**Latent dynamics**

Let $\mathbf{x}(t)$ denote a (vector-valued) latent process, where each component $x_j(t)$, $j \in (1, \ldots, P)$, evolves according to an independent Gaussian process (GP),

$$x_j(t) \sim \mathcal{GP}\left(0, k_t\right), \tag{1}$$

with covariance function $k_t(t, t') \triangleq \text{cov}(x_j(t), x_j(t'))$ governing how each scalar process varies over time. Although we can select any valid covariance function for $k_t$, here we use the exponential covariance function, a special case of the Matérn kernel, given by $k(t, t') = r \exp\left(-|t - t'|/l\right)$, which is parametrized by a marginal variance $r > 0$ and length-scale $l > 0$. Samples from this GP are continuous but not differentiable, equivalent to a Gaussian random walk with a bias toward the origin, also known as the Ornstein-Uhlenbeck process [17].

The latent state $\mathbf{x}(t)$ at any time $t$ is a $P$-dimensional vector that we will write as $\mathbf{x}_t \in \mathbb{R}^{P \times 1}$. The collection of such vectors over $T$ time bins forms a matrix $\mathbf{X} \in \mathbb{R}^{P \times T}$. Let $\mathbf{x}_j$ denote the $j$th row of $\mathbf{X}$, which contains the set of states in latent dimension $j$. From the definition of a GP, $\mathbf{x}_j$ has a multivariate normal distribution,

$$\mathbf{x}_j \sim \mathcal{N}(\mathbf{0}, K_t) \tag{2}$$

with a $T \times T$ covariance matrix $K_t$ generated by evaluating the covariance function $k_t$ at all time bins in $(1, \ldots, T)$.

**Nonlinear mapping**

Let $h : \mathbb{R}^P \longrightarrow \mathbb{R}$ denote a nonlinear function mapping from the latent vector $\mathbf{x}_t$ to a firing rate $\lambda_t$. We will refer to $h(\mathbf{x})$ as a tuning curve, although unlike traditional tuning curves, which describe firing rate as a function of some externally (observable) stimulus parameter, here $h(\mathbf{x})$ describes firing rate as a function of the (unobserved) latent vector $\mathbf{x}$. Previous work has modeled $h$ with a parametric nonlinear function such as a deep neural network [9, 10]. Here we develop a nonparametric approach using a Gaussian process prior over the log of $h$. The logarithm assures that spike rates are non-negative.

Let $f_i(\mathbf{x}) = \log h_i(\mathbf{x})$ denote the log tuning curve for the $i$'th neuron in our population, which we model with a GP,

$$f_i(\mathbf{x}) \sim \mathcal{GP}\left(0, k_x\right), \tag{3}$$

where $k_x$ is a (spatial) covariance function that governs smoothness of the function over its $P$-dimensional input space. For simplicity, we use the common Gaussian or radial basis function (RBF) covariance function: $k_x(\mathbf{x}, \mathbf{x}') = \rho \exp\left(-||\mathbf{x} - \mathbf{x}'||_2^2 / 2\delta^2\right)$, where $\mathbf{x}$ and $\mathbf{x}'$ are arbitrary points in latent space, $\rho$ is the marginal variance and $\delta$ is the length scale. The tuning curve for neuron $i$ is then given by $h_i(\mathbf{x}) = \exp(f_i(\mathbf{x}))$.

Let $\mathbf{f}_i \in \mathbb{R}^{T \times 1}$ denote a vector with the $t$'th element equal to $f_i(\mathbf{x}_t)$. From the definition of a GP, $\mathbf{f}_i$ has a multivariate normal distribution given latent vectors at all time bins $\mathbf{x}_{1:T} = \{\mathbf{x}_t\}_{t=1}^T$,

$$\mathbf{f}_i | \mathbf{x}_{1:T} \sim \mathcal{N}(\mathbf{0}, K_x) \tag{4}$$

with a $T \times T$ covariance matrix $K_x$ generated by evaluating the covariance function $k_x$ at all pairs of latent vectors in $\mathbf{x}_{1:T}$. Stacking $\mathbf{f}_i$ for $N$ neurons, we will formulate a matrix $\mathbf{F} \in \mathbb{R}^{N \times T}$ with $\mathbf{f}_i^\top$ on the $i$'th row. The element on the $i$'th row and the $t$'th column is $f_{i,t} = f_i(\mathbf{x}_t)$.

**Poisson spiking**

Lastly, we assume Poisson spiking given the latent firing rates. We assume that spike rates are in units of spikes per time bin. Let $\lambda_{i,t} = \exp(f_{i,t}) = \exp(f_i(\mathbf{x}_t))$ denote the spike rate of neuron $i$ at time $t$. The spike-count of neuron $i$ at $t$ given the log tuning curve $f_i$ and latent vector $\mathbf{x}_t$ is Poisson distributed as

$$y_{i,t} | f_i, \mathbf{x}_t \sim \text{Poiss}(\exp(f_i(\mathbf{x}_t))). \tag{5}$$

In summary, our model is as a doubly nonlinear Gaussian process latent variable model with Poisson observations (P-GPLVM). One GP is used to model the nonlinear evolution of the latent dynamic $x$, while a second GP is used to generate the log of the tuning curve $f$ as a nonlinear function of $x$, which is then mapped to a tuning curve $h$ via a nonlinear link function, e.g. exponential function. Fig. 1 provides a schematic of the model.

# 3 Inference using the decoupled Laplace approximation

For our inference procedure, we estimate the log of the tuning curve, $f$, as opposed to attempting to infer the tuning curve $h$ directly. Once $f$ is estimated, $h$ can be obtained by exponentiating $f$. Given the model outlined above, the joint distribution over the observed data and all latent variables is written as,

$$p(\mathbf{Y}, \mathbf{F}, \mathbf{X}, \boldsymbol{\theta}) = p(\mathbf{Y}|\mathbf{F})p(\mathbf{F}|\mathbf{X}, \rho, \delta)p(\mathbf{X}|r, l) = \prod_{i=1}^{N}\prod_{t=1}^{T} p(y_{i,t}|f_{i,t}) \prod_{i=1}^{N} p(\mathbf{f}_i|\mathbf{X}, \rho, \delta) \prod_{j=1}^{P} p(\mathbf{x}_j|r, l), \quad (6)$$

where $\boldsymbol{\theta} = \{\rho, \delta, r, l\}$ is the hyperparameter set, references to which will now be suppressed for simplification. This is a Gaussian process latent variable model (GPLVM) with Poisson observations and a GP prior, and our goal is to now estimate both $\mathbf{F}$ and $\mathbf{X}$. A standard Bayesian treatment of the GPLVM requires the computation of the log marginal likelihood associated with the joint distribution (Eq.6). Both $\mathbf{F}$ and $\mathbf{X}$ must be marginalized out,

$$\log p(\mathbf{Y}) = \log \int \int p(\mathbf{Y}, \mathbf{F}, \mathbf{X})d\mathbf{X}d\mathbf{F} = \log \int p(\mathbf{Y}|\mathbf{F}) \int p(\mathbf{F}|\mathbf{X})p(\mathbf{X})d\mathbf{X} \ d\mathbf{F}. \quad (7)$$

However, propagating the prior density $p(\mathbf{X})$ through the nonlinear mapping makes this inference difficult. The nested integral in (Eq. 7) contains $\mathbf{X}$ in a complex nonlinear manner, making analytical integration over $\mathbf{X}$ infeasible. To overcome these difficulties, we can use a straightforward MAP training procedure where the latent variables $\mathbf{F}$ and $\mathbf{X}$ are selected according to

$$\mathbf{F}_{\text{MAP}}, \mathbf{X}_{\text{MAP}} = \text{argmax}_{\mathbf{F}, \mathbf{X}} \ p(\mathbf{Y}|\mathbf{F})p(\mathbf{F}|\mathbf{X})p(\mathbf{X}). \quad (8)$$

Note that point estimates of the hyperparameters $\boldsymbol{\theta}$ can also be found by maximizing the same objective function. As discussed above, learning $\mathbf{X}$ remains a challenge due to the interplay of the latent variables, i.e. the dependency of $\mathbf{F}$ on $\mathbf{X}$. For our MAP training procedure, fixing one latent variable while optimizing for the other in a coordinate descent approach is highly inefficient since the strong interplay of variables often means getting trapped in bad local optima. In variational GPLVM [18], the authors introduced a non-standard variational inference framework for approximately integrating out the latent variables $\mathbf{X}$ then subsequently training a GPLVM by maximizing an analytic lower bound on the exact marginal likelihood. An advantage of the variational framework is the introduction of auxiliary variables which weaken the strong dependency between $\mathbf{X}$ and $\mathbf{F}$. However, the variational approximation is only applicable to Gaussian observations; with Poisson observations, the integral over $\mathbf{F}$ remains intractable. In the following, we will propose using variations of the Laplace approximation for inference.

## 3.1 Standard Laplace approximation

We first use Laplace's method to find a Gaussian approximation $q(\mathbf{F}|\mathbf{Y}, \mathbf{X})$ to the true posterior $p(\mathbf{F}|\mathbf{Y}, \mathbf{X})$, then do MAP estimation for $\mathbf{X}$ only. We employ the Laplace approximation for each $\mathbf{f}_i$ individually. Doing a second order Taylor expansion of $\log p(\mathbf{f}_i|\mathbf{y}_i, \mathbf{X})$ around the maximum of the posterior, we obtain a Gaussian approximation

$$q(\mathbf{f}_i|\mathbf{y}_i, \mathbf{X}) = \mathcal{N}(\hat{\mathbf{f}}_i, A^{-1}), \quad (9)$$

where $\hat{\mathbf{f}}_i = \text{argmax}_{\mathbf{f}_i} \ p(\mathbf{f}_i|\mathbf{y}_i, \mathbf{X})$ and $A = -\nabla\nabla \log p(\mathbf{f}_i|\mathbf{y}_i, \mathbf{X})|_{\mathbf{f}_i=\hat{\mathbf{f}}_i}$ is the Hessian of the negative log posterior at that point. By Bayes' rule, the posterior over $\mathbf{f}_i$ is given by $p(\mathbf{f}_i|\mathbf{y}_i, \mathbf{X}) = p(\mathbf{y}_i|\mathbf{f}_i)p(\mathbf{f}_i|\mathbf{X})/p(\mathbf{y}_i|\mathbf{X})$, but since $p(\mathbf{y}_i|\mathbf{X})$ is independent of $\mathbf{f}_i$, we need only consider the unnormalized posterior, defined as $\Psi(\mathbf{f}_i)$, when maximizing w.r.t. $\mathbf{f}_i$. Taking the logarithm gives

$$\Psi(\mathbf{f}_i) = \log p(\mathbf{y}_i|\mathbf{f}_i) + \log p(\mathbf{f}_i|\mathbf{X}) = \log p(\mathbf{y}_i|\mathbf{f}_i) - \frac{1}{2}\mathbf{f}_i^\top K_x^{-1}\mathbf{f}_i - \frac{1}{2}\log|K_x| + const. \quad (10)$$

Differentiating (Eq. 10) w.r.t. $\mathbf{f}_i$ we obtain

$$\nabla\Psi(\mathbf{f}_i) = \nabla \log p(\mathbf{y}_i|\mathbf{f}_i) - K_x^{-1}\mathbf{f}_i \quad (11)$$

$$\nabla\nabla\Psi(\mathbf{f}_i) = \nabla\nabla \log p(\mathbf{y}_i|\mathbf{f}_i) - K_x^{-1} = -W_i - K_x^{-1}, \quad (12)$$

where $W_i = -\nabla\nabla \log p(\mathbf{y}_i|\mathbf{f}_i)$. The approximated log conditional likelihood on $\mathbf{X}$ (see Sec. 3.4.4 in [17]) can then be written as

$$\log q(\mathbf{y}_i|\mathbf{X}) = \log p(\mathbf{y}_i|\hat{\mathbf{f}}_i) - \frac{1}{2}\hat{\mathbf{f}}_i^\top K_x^{-1}\hat{\mathbf{f}}_i - \frac{1}{2}\log|I_T + K_xW_i|. \quad (13)$$

We can then estimate $\mathbf{X}$ as

$$\mathbf{X}_{\text{MAP}} = \text{argmax}_{\mathbf{X}} \sum_{i=1}^{N} q(\mathbf{y}_i|\mathbf{X})p(\mathbf{X}). \tag{14}$$

When using standard LA, the gradient of $\log q(\mathbf{y}_i|\mathbf{X})$ w.r.t. $\mathbf{X}$ should be calculated for a given posterior mode $\hat{\mathbf{f}}_i$. Note that not only is the covariance matrix $K_x$ an explicit function of $\mathbf{X}$, but also $\hat{\mathbf{f}}_i$ and $W_i$ are also implicitly functions of $\mathbf{X}$ — when $\mathbf{X}$ changes, the optimum of the posterior $\hat{\mathbf{f}}_i$ changes as well. Therefore, $\log q(\mathbf{y}_i|\mathbf{X})$ contains an implicit function of $\mathbf{X}$ which does not allow for a straightforward closed-form gradient expression. Calculating numerical gradients instead yields a very inefficient implementation empirically.

## 3.2 Third-derivative Laplace approximation

One method to derive this gradient explicitly is described in [17] (see Sec. 5.5.1). We adapt their procedure to our setting to make the implicit dependency of $\hat{\mathbf{f}}_i$ and $W_i$ on $\mathbf{X}$ explicit. To solve (Eq. 14), we need to determine the partial derivative of our approximated log conditional likelihood (Eq. 13) w.r.t. $\mathbf{X}$, given as

$$\frac{\partial \log q(\mathbf{y}_i|\mathbf{X})}{\partial \mathbf{X}} = \left. \frac{\partial \log q(\mathbf{y}_i|\mathbf{X})}{\partial \mathbf{X}} \right|_{\text{explicit}} + \sum_{t=1}^{T} \frac{\partial \log q(\mathbf{y}_i|\mathbf{X})}{\partial \hat{f}_{i,t}} \frac{\partial \hat{f}_{i,t}}{\partial \mathbf{X}} \tag{15}$$

by the chain rule. When evaluating the second term, we use the fact that $\hat{\mathbf{f}}_i$ is the posterior maximum, so $\partial \Psi(\mathbf{f}_i)/\partial \mathbf{f}_i = \mathbf{0}$ at $\mathbf{f}_i = \hat{\mathbf{f}}_i$, where $\Psi(\mathbf{f}_i)$ is defined in (Eq. 11). Thus the implicit derivatives of the first two terms in (Eq. 13) vanish, leaving only

$$\frac{\partial \log q(\mathbf{y}_i|\mathbf{X})}{\partial \hat{f}_{i,t}} = -\frac{1}{2}\text{tr}\left( (K_x^{-1} + W_i)^{-1} \frac{\partial W_i}{\partial \hat{f}_{i,t}} \right) = -\frac{1}{2}\left[ (K_x^{-1} + W_i)^{-1} \right]_{tt} \frac{\partial^3}{\partial \hat{f}_{i,t}^3} \log p(\mathbf{y}_i|\hat{\mathbf{f}}_i). \tag{16}$$

To evaluate $\partial \hat{f}_{i,t}/\partial \mathbf{X}$, we differentiate the self-consistent equation $\hat{\mathbf{f}}_i = K_x \nabla \log p(\mathbf{y}_i|\hat{\mathbf{f}}_i)$ (setting (Eq. 11) to be 0 at $\hat{\mathbf{f}}_i$) to obtain

$$\frac{\partial \hat{\mathbf{f}}_i}{\partial \mathbf{X}} = \frac{\partial K_x}{\partial \mathbf{X}} \nabla \log p(\mathbf{y}_i|\hat{\mathbf{f}}_i) + K_x \frac{\nabla \log p(\mathbf{y}_i|\hat{\mathbf{f}}_i)}{\partial \hat{\mathbf{f}}_i} \frac{\partial \hat{\mathbf{f}}_i}{\partial \mathbf{X}} = (I_T + K_x W_i)^{-1} \frac{\partial K_x}{\partial \mathbf{X}} \nabla \log p(\mathbf{y}_i|\hat{\mathbf{f}}_i), \tag{17}$$

where we use the chain rule $\frac{\partial}{\partial \mathbf{X}} = \frac{\partial \hat{\mathbf{f}}_i}{\partial \mathbf{X}} \cdot \frac{\partial}{\partial \hat{\mathbf{f}}_i}$ and $\partial \nabla \log p(\mathbf{y}_i|\hat{\mathbf{f}}_i)/\partial \hat{\mathbf{f}}_i = -W_i$ from (Eq. 12). The desired implicit derivative is obtained by multiplying (Eq. 16) and (Eq. 17) to formulate the second term in (Eq. 15).

We can now estimate $\mathbf{X}_{\text{MAP}}$ with (Eq. 14) using the explicit gradient expression in (Eq. 15). We call this method *third-derivative Laplace approximation* (tLA), as it depends on the third derivative of the data likelihood term (see [17] for further details). However, there is a big computational drawback with tLA: for each step along the gradient we have just derived, the posterior mode $\hat{\mathbf{f}}_i$ must be reevaluated. This method might lead to a fast convergence theoretically, but this nested optimization makes for a very slow computation empirically.

## 3.3 Decoupled Laplace approximation

We propose a novel method to relax the Laplace approximation, which we refer to as the *decoupled Laplace approximation* (dLA). Our relaxation not only decouples the strong dependency between $\mathbf{X}$ and $\mathbf{F}$, but also avoids the nested optimization of searching for the posterior mode of $\mathbf{F}$ within each update of $\mathbf{X}$. As in tLA, dLA also assumes $\hat{\mathbf{f}}_i$ to be a function of $\mathbf{X}$. However, while tLA assumes $\hat{\mathbf{f}}_i$ to be an implicit function of $\mathbf{X}$, dLA constructs an explicit mapping between $\hat{\mathbf{f}}_i$ and $\mathbf{X}$.

The standard Laplace approximation uses a Gaussian approximation for the posterior $p(\mathbf{f}_i|\mathbf{y}_i, \mathbf{X}) \propto p(\mathbf{y}_i|\mathbf{f}_i)p(\mathbf{f}_i|\mathbf{X})$ where, in this paper, $p(\mathbf{y}_i|\mathbf{f}_i)$ is a Poisson distribution and $p(\mathbf{f}_i|\mathbf{X})$ is a multivariate Gaussian distribution. We first do the same second order Taylor expansion of $\log p(\mathbf{f}_i|\mathbf{y}_i, \mathbf{X})$ around the posterior maximum to find $q(\mathbf{f}_i|\mathbf{y}_i, \mathbf{X})$ as in (Eq. 9). Now if we approximate the likelihood distribution $p(\mathbf{y}_i|\mathbf{f}_i)$ as a Gaussian distribution $q(\mathbf{y}_i|\mathbf{f}_i) = \mathcal{N}(\mathbf{m}, S)$, we can derive its mean $\mathbf{m}$ and covariance $S$. If $p(\mathbf{f}_i|\mathbf{X}) = \mathcal{N}(\mathbf{0}, K_x)$ and $q(\mathbf{f}_i|\mathbf{y}_i, \mathbf{X}) = \mathcal{N}(\hat{\mathbf{f}}_i, A^{-1})$, the relationship between two Gaussian distributions and their product allow us to solve for $\mathbf{m}$ and $S$ from the relationship $\mathcal{N}(\hat{\mathbf{f}}_i, A^{-1}) \propto \mathcal{N}(\mathbf{m}, S)\mathcal{N}(\mathbf{0}, K_x)$:

---
**Algorithm 1** Decoupled Laplace approximation at iteration $k$
---
**Input:** data observation $\mathbf{y}_i$, latent variable $\mathbf{X}^{k-1}$ from iteration $k-1$

1. Compute the new posterior mode $\hat{\mathbf{f}}_i^k$ and the precision matrix $A^k$ by solving (Eq. 10) to obtain $q(\mathbf{f}_i|\mathbf{y}_i, \mathbf{X}^{k-1}) = \mathcal{N}(\hat{\mathbf{f}}_i^k, A^{k^{-1}})$.

2. Derive $\mathbf{m}^k$ and $S^k$ (Eq. 18): $S^k = (A^k - K_x^{-1})^{-1}$, $\mathbf{m}^k = S^k A^k \hat{\mathbf{f}}_i^k$.

3. Fix $\mathbf{m}^k$ and $S^k$ and derive the new mean and covariance for $q(\mathbf{f}_i|\mathbf{y}_i, \mathbf{X}^{k-1})$ as functions of $\mathbf{X}$: $A(\mathbf{X}) = S^{k^{-1}} + K_x(\mathbf{X})^{-1}$, $\hat{\mathbf{f}}_i(\mathbf{X}) = A(\mathbf{X})^{-1} S^{k^{-1}} \mathbf{m}^k = A(\mathbf{X})^{-1} A^k \hat{\mathbf{f}}_i^k$.

4. Since $A = W_i + K_x^{-1}$, we have $W_i = S^{k^{-1}}$, and can obtain the new approximated conditional distribution $q(\mathbf{y}_i|\mathbf{X})$ (Eq. 13) with $\hat{\mathbf{f}}_i$ replaced by $\hat{\mathbf{f}}_i(\mathbf{X})$.

5. Solve $\mathbf{X}^k = \text{argmax}_\mathbf{X} \sum_{i=1}^N q(\mathbf{y}_i|\mathbf{X})p(\mathbf{X})$.

**Output:** new latent variable $\mathbf{X}^k$
---

$$A = S^{-1} + K_x^{-1}, \; \hat{\mathbf{f}}_i = A^{-1}S^{-1}\mathbf{m} \quad \Longrightarrow \quad S = (A - K_x^{-1})^{-1}, \; \mathbf{m} = SA\hat{\mathbf{f}}_i. \tag{18}$$

$\mathbf{m}$ and $S$ represent the components of the posterior terms, $\hat{\mathbf{f}}_i$ and $A$, that come from the likelihood. Now when estimating $\mathbf{X}$, we fix these likelihood terms $\mathbf{m}$ and $S$, and completely relax the prior, $p(\mathbf{f}_i|\mathbf{X})$. We are still solving (Eq. 14) w.r.t. $\mathbf{X}$, but now $q(\mathbf{f}_i|\mathbf{y}_i, \mathbf{X})$ has both mean and covariance approximated as explicit functions of $\mathbf{X}$. Alg. 1 describes iteration $k$ of the dLA algorithm, with which we can now estimate $\mathbf{X}_{\text{MAP}}$. Step 3 indicates that the posterior maximum for the current iteration $\hat{\mathbf{f}}_i(\mathbf{X}) = A(\mathbf{X})^{-1} A^k \hat{\mathbf{f}}_i^k$ is now explicitly updated as a function of $\mathbf{X}$, avoiding the computationally demanding nested optimization of tLA. Intuitively, dLA works by finding a Gaussian approximation to the likelihood at $\hat{\mathbf{f}}_i^k$ such that the approximated posterior of $\mathbf{f}_i$, $q(\mathbf{f}_i|\mathbf{y}_i, \mathbf{X})$, is now a closed-form Gaussian distribution with mean and covariance as functions of $\mathbf{X}$, ultimately allowing for the explicit calculation of $q(\mathbf{y}_i|\mathbf{X})$.

## 4 Tuning curve estimation

Given the estimated $\hat{\mathbf{X}}$ and $\hat{\mathbf{f}}$ from the inference, we can now calculate the tuning curve $h$ for each neuron. Let $\mathbf{x}_{1:G} = \{\mathbf{x}_g\}_{g=1}^G$ be a grid of $G$ latent states, where $\mathbf{x}_g \in \mathbb{R}^{P \times 1}$. Correspondingly, for each neuron, we have the log of the tuning curve vector evaluated on the grid of latent states, $\mathbf{f}_{\text{grid}} \in \mathbb{R}^{G \times 1}$, with the $g$'th element equal to $f(\mathbf{x}_g)$. Similar to (Eq. 4), we can write down its distribution as

$$\mathbf{f}_{\text{grid}}|\mathbf{x}_{1:G} \sim \mathcal{N}(\mathbf{0}, K_{\text{grid}}) \tag{19}$$

with a $G \times G$ covariance matrix $K_{\text{grid}}$ generated by evaluating the covariance function $k_x$ at all pairs of vectors in $\mathbf{x}_{1:G}$. Therefore we can write a joint distribution for $[\hat{\mathbf{f}}, \mathbf{f}_{\text{grid}}]$ as

$$\begin{bmatrix} \hat{\mathbf{f}} \\ \mathbf{f}_{\text{grid}} \end{bmatrix} \sim \mathcal{N}\left(\mathbf{0}, \begin{bmatrix} K_{\hat{\mathbf{x}}} & \mathbf{k}_{\text{grid}} \\ \mathbf{k}_{\text{grid}}^\top & K_{\text{grid}} \end{bmatrix}\right). \tag{20}$$

$K_{\hat{\mathbf{x}}} \in \mathbb{R}^{T \times T}$ is a covariance matrix with elements evaluated at all pairs of estimated latent vectors $\hat{\mathbf{x}}_{1:T} = \{\hat{\mathbf{x}}_t\}_{t=1}^T$ in $\hat{\mathbf{X}}$, and $\mathbf{k}_{\text{grid}_{t,g}} = k_x(\hat{\mathbf{x}}_t, \mathbf{x}_g)$. Thus we have the following posterior distribution over $\mathbf{f}_{\text{grid}}$:

$$\mathbf{f}_{\text{grid}}|\hat{\mathbf{f}}, \hat{\mathbf{x}}_{1:T}, \mathbf{x}_{1:G} \quad \sim \quad \mathcal{N}(\mu(\mathbf{x}_{1:G}), \Sigma(\mathbf{x}_{1:G})) \tag{21}$$
$$\mu(\mathbf{x}_{1:G}) = \mathbf{k}_{\text{grid}}^\top K_{\hat{\mathbf{x}}}^{-1}\hat{\mathbf{f}} \quad , \quad \Sigma(\mathbf{x}_{1:G}) = \text{diag}(K_{\text{grid}}) - \mathbf{k}_{\text{grid}}^\top K_{\hat{\mathbf{x}}}^{-1}\mathbf{k}_{\text{grid}}$$

where $\text{diag}(K_{\text{grid}})$ denotes a diagonal matrix constructed from the diagonal of $K_{\text{grid}}$. Setting $\hat{\mathbf{f}}_{\text{grid}} = \mu(\mathbf{x}_{1:G})$, the spike rate vector

$$\hat{\boldsymbol{\lambda}}_{\text{grid}} = \exp(\hat{\mathbf{f}}_{\text{grid}}) \tag{22}$$

describes the tuning curve $h$ evaluated on the grid $\mathbf{x}_{1:G}$.

## 5 Experiments

### 5.1 Simulation data

We first examine performance using two simulated datasets generated with different kinds of tuning curves, namely sinusoids and Gaussian bumps. We will compare our algorithm (P-GPLVM) with

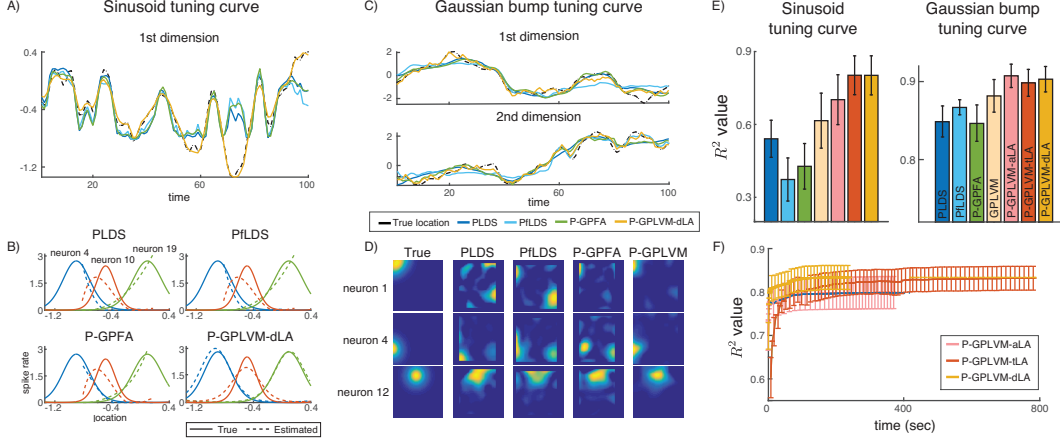

Figure 2: Results from the sinusoid and Gaussian bump simulated experiments. A) and C) are estimated latent processes. B) and D) display the tuning curves estimated by different methods. E) shows the $R^2$ performances with error bars. F) shows the convergence $R^2$ performances of three different Laplace approximation inference methods with error bars. Error bars are plotted every 10 seconds.

PLDS, PfLDS, P-GPFA and GPLVM (see Table 1), using the tLA and dLA inference methods. We also include an additional variant on the Laplace approximation, which we call the *approximated Laplace approximation* (aLA), where we use only the explicit (first) term in (Eq. 15) to optimize over $\mathbf{X}$ for multiple steps given a fixed $\hat{\mathbf{f}}_i$. This allows for a coarse estimation for the gradient w.r.t. $\mathbf{X}$ for a few steps in $\mathbf{X}$ before estimation is necessary, partially relaxing the nested optimization so as to speed up the learning procedure.

For comparison between models in our simulated experiments, we compute the R-squared ($R^2$) values from the known latent processes and the estimated latent processes. In all simulation studies, we generate 1 single trial per neuron with 20 simulated neurons and 100 time bins for a single experiment. Each experiment is repeated 10 times and results are averaged across 10 repeats.

**Sinusoid tuning curve:** This simulation generates a "grid cell" type response. A grid cell is a type of neuron that is activated when an animal occupies any point on a grid spanning the environment [19]. When an animal moves in a one-dimensional space ($P = 1$), grid cells exhibit oscillatory responses. Motivated by the response properties of grid cells, the log firing rate of each neuron $i$ is coupled to the latent process through a sinusoid with a neuron-specific phase $\Phi_i$ and frequency $\omega_i$,

$$f_i = \sin(\omega_i x + \Phi_i). \tag{23}$$

We randomly generated $\Phi_i$ uniformly from the region $[0, 2\pi]$ and $\omega_i$ uniformly from $[1.0, 4.0]$.

An example of the estimated latent processes versus the true latent process is presented in Fig. 2A. We used least-square regression to learn an affine transformation from the latent space to the space of the true locations. Only P-GPLVM finds the global optimum by fitting the valley around $t = 70$. Fig. 2B displays the true tuning curves and the estimated tuning curves for neuron 4, 10, & 9 with PLDS, PfLDS, P-GPFA and P-GPLVM-dLA. For PLDS, PfLDS and P-GPFA, we replace the estimated $\hat{\mathbf{f}}$ with the observed spike count $\mathbf{y}$ in (Eq. 21), and treat the posterior mean as the tuning curve on a grid of latent representations. For P-GPLVM, the tuning curve is estimated via (Eq. 22). The $R^2$ performance is shown in the first column of Fig. 2E.

**Deterministic Gaussian bump tuning curve:** For this simulation, each neuron's tuning curve is modeled as a unimodal Gaussian bump in a 2D space such that the log of the tuning curve, $f$, is a deterministic Gaussian function of $x$. Fig. 2C shows an example of the estimated latent processes. PLDS fits an overly smooth curve, while P-GPLVM can find the small wiggles that are missed by other methods. Fig. 2D displays the 2D tuning curves for neuron 1, 4, & 12 estimated by PLDS, PfLDS, P-GPFA and P-GPLVM-dLA. The $R^2$ performance is shown in the second column of Fig. 2E.

Overall, P-GPFA has a quite unstable performance due to the ARD kernel function in the GP prior, potentially encouraging a bias for smoothness even when the underlying latent process is actually

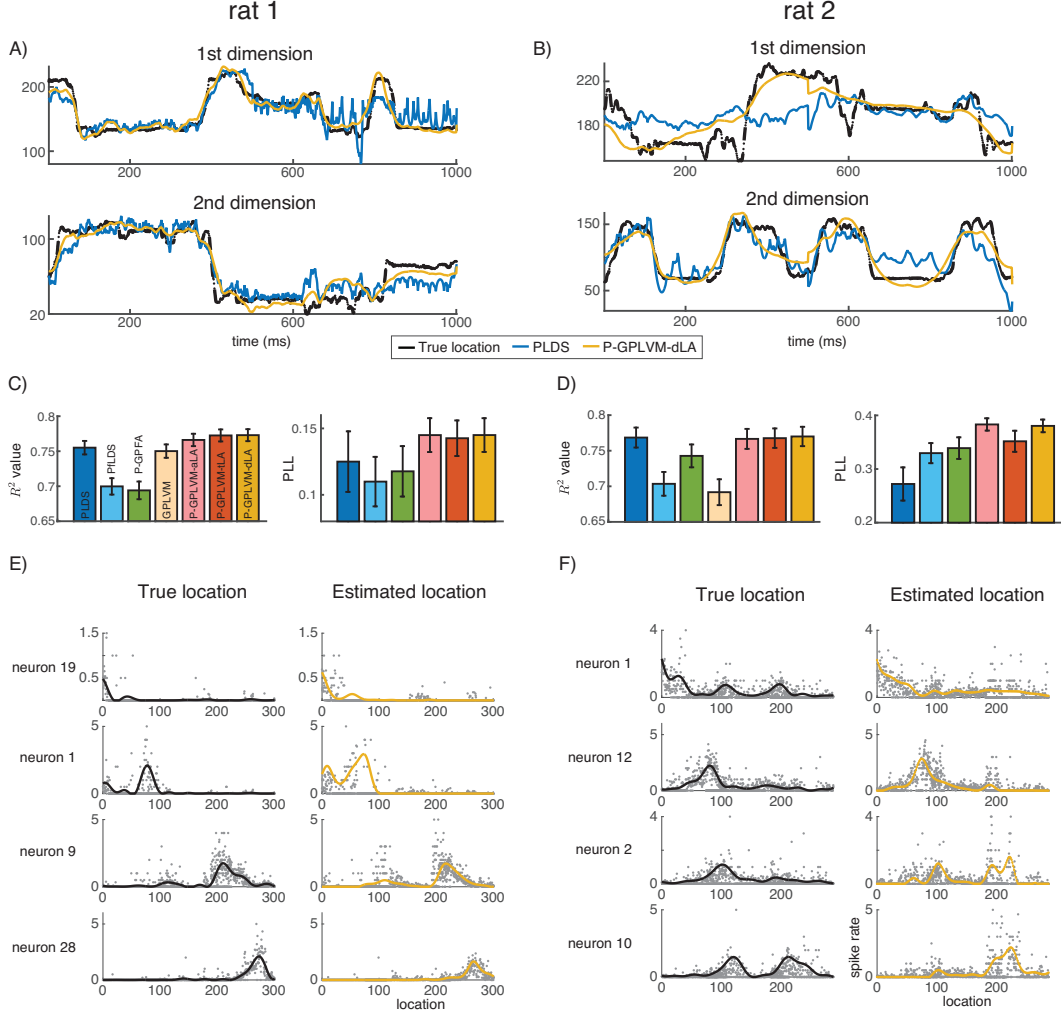

Figure 3: Results from the hippocampal data of two rats. A) and B) are estimated latent processes during a 1s recording period for two rats. C) and D) show $R^2$ and PLL performance with error bars. E) and F) display the true tuning curves and the tuning curves estimated by P-GPLVM-dLA.

quite non-smooth. PfLDS performs better than PLDS in the second case, but when the true latent process is highly nonlinear (sinusoid) and the single-trial dataset is small, PfLDS losses its advantage to stochastic optimization. GPLVM has a reasonably good performance with the nonlinearities, but is worse than P-GPLVM which demonstrates the significance of using the Poisson observation model. For P-GPLVM, the dLA inference algorithm performs best overall w.r.t. both convergence speed and $R^2$ (Fig. 2F).

## 5.2 Application to rat hippocampal neuron data

Next, we apply the proposed methods to extracellular recordings from the rodent hippocampus. Neurons were recorded bilaterally from the pyramidal layer of CA3 and CA1 in two rats as they performed a spatial alternation task on a W-shaped maze [20]. We confine our analyses to simultaneously recorded putative place cells during times of active navigation. Total number of simultaneously recorded neurons ranged from 7-19 for rat 1 and 24-38 for rat 2. Individual trials of 50 seconds were isolated from 15 minute recordings, and binned at a resolution of 100ms.

We used this hippocampal data to identify a 2D latent space using PLDS, PfLDS, P-GPFA, GPLVM and P-GPLVMs (Fig. 3), and compared these to the true 2D location of the rodent. For visualization purposes, we linearized the coordinates along the arms of the maze to obtain 1D representations.

Fig. 3A & B present two segments of 1s recordings for the two animals. The P-GPLVM results are smoother and recover short time-scale variations that PLDS ignores. The average $R^2$ performance for all methods for each rodent is shown in Fig. 3C & D where P-GPLVM-dLA consistently performs the best.

We also assessed the model fitting quality by doing prediction on a held-out dataset. We split all the time bins in each trial into training time bins (the first 90% time bins) and held-out time bins (the last 10% time bins). We first estimated the parameters for the mapping function or the tuning curve in each model using spike trains from all the neurons within training time bins. Then we fixed the parameters and inferred the latent process using spike trains from 70% neurons within held-out time bins. Finally, we calculated the predictive log likelihood (PLL) for the other 30% neurons within held-out time bins given the inferred latent process. We subtracted the log-likelihood of the population mean firing rate model (single spike rate) from the predictive log likelihood divided by number of observations, shown in Fig. 3C & D. Both P-GPLVM-aLA and P-GPLVM-dLA perform well. GPLVM has very negative PLL, omitted in the figures.

Fig. 3E & F present the tuning curves learned by P-GPLVM-dLA where each row corresponds to a neuron. For our analysis we have the true locations $\mathbf{x}_{\text{true}}$, the estimated locations $\mathbf{x}_{\text{P-GPLVM}}$, a grid of $G$ locations $\mathbf{x}_{1:G}$ distributed with a shape of the maze, the spike count observation $\mathbf{y}_i$, and the estimated log of the tuning curves $\hat{\mathbf{f}}_i$ for each neuron $i$. The light gray dots in the first column of Fig. 3E & F are the binned spike counts when mapping from the space of $\mathbf{x}_{\text{true}}$ to the space of $\mathbf{x}_{1:G}$. The second column contains the binned spike counts mapped from the space of $\mathbf{x}_{\text{P-GPLVM}}$ to the space of $\mathbf{x}_{1:G}$. The black curves in the first column are achieved by replacing $\hat{\mathbf{x}}$ and $\hat{\mathbf{f}}$ with $\mathbf{x}_{\text{true}}$ and $\mathbf{y}$ respectively using the predictive posterior in (Eq. 21) and (Eq. 22). The yellow curves in the second column are the estimated tuning curves by using (Eq. 22) to get $\hat{\boldsymbol{\lambda}}_{\text{grid}}$ for each neuron. We can tell that the estimated tuning curves closely match the true tuning curves from the observations, discovering different responsive locations for different neurons as the rat moves.

# 6 Conclusion

We proposed a doubly nonlinear Gaussian process latent variable model for neural population spike trains that can identify nonlinear low-dimensional structure underlying apparently high-dimensional spike train data. We also introduced a novel decoupled Laplace approximation, a fast approximate inference method that allows us to efficiently maximize marginal likelihood for the latent path while integrating over tuning curves. We showed that this method outperforms previous Laplace-approximation-based inference methods in both the speed of convergence and accuracy. We applied the model to both simulated data and spike trains recorded from hippocampal place cells and showed that it outperforms a variety of previous methods for latent structure discovery.

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
