[Reviews · NeurIPS 2017]

Reviewer 1



I was not expecting to read a paper claiming state-of-the-art results with GPLVM in 2017, but here we are. The model is a very solid application of nonparametric inference, including a novel extension of the Laplace approximation appropriate for GPLVMs with Poisson observations, and the results are interesting in that they seem better able to decode known behaviorally-relevant variables in the one real dataset they look at. A few objections/questions: -In the abstract, the authors claim their results are superior to those of variational autoencoder based models like LFADS. However, no experimental results are presented. Did the authors forget to remove this claim from the abstract? The authors do note that GP-based models tend to outperform deep neural networks in the small data limit, which seems accurate, but some experimental comparison would be appreciated. And given the constantly growing volume of neural data, a comparison on larger datasets would be appreciated as well. -Experimental results are judged entirely based on r^2 scores with a known behavioral variable. While this may be of more interest for neuroscientists, a proper comparison of held-out data likelihood or approximate marginal likelihood of the data would be appreciated. -In the experiments with hippocampal data, how are the latent variables aligned with behavioral variables? Linear regression? Or did the latent variables discovered just happen to align perfectly with x and y directions in the maze? The latter seems highly unlikely.

Reviewer 2



Here the authors introduce a latent variable model for population neural activity with GP latent state and, also, GP mapping (tuning curves) between the latent state and Poisson observations. This is an interesting model that nicely extends work in this area. I have a couple points of concern… Identifiability – if I understand correctly rotations and reflections of x’=Rx have the same likelihood. How can the latent variables that the model estimates be directly compared to extrinsic variables in this situation. Why does r^2 work? In general, why should the estimated latent variables match the extrinsic location one-to-one? In other contexts (e.g. Vidne et al. J Comp Neuro 2012), the low-dimensional latent state has been interpreted as common input, for instance, that can be completely unrelated to some covariate. I’m also a bit confused how the latent variables from PLDS and P-GPFA can be directly compared with those from this new method, since the mapping function is linear for those methods. In Fig 2, where are the tuning curves for PLDS/P-GPFA coming from, for instance? Minor issues: In section 2/3 – it’s mostly clear from the context, but you may want to explicitly state that \rho and \delta are the marginal variance and length scale for k_x. It could be argued that the task dimensionality is often unclear. Even in hippocampus, phenomena such as head-direction cells, speed tuning, and phase locking would suggest that neural populations are acting in more than 2d. ** After author comments ** The authors have addressed all of my concerns. Using regression to map the latent variables to the extrinsic variables is a nice approach.

Reviewer 3



Summary --------- The authors propose a gaussian process latent variable model with poisson observations (P-GPLVM). The demonstrate how to perform inference using a decoupled Laplace approximation. They apply the model to simulated data as well as rat hippocampal data. The paper is well written and clearly presented. I appreciated the nice summary of the model structures in Table 1. I would have liked to see more applications of the technique, and/or a larger improvement on real data compared to previous methods. Questions/concerns -------------------- - The performance benefit of the P-GPLVM is quite small over PLDS and the GPLVM on real data. For example, in Figure 2C, there is only a weak improvement over these methods (especially given the fact that the y-scale in Fig 2C shows a small range, and lack error bars), lessening the overall impact of the results. - How does the algorithm scale with the size of the latent dimension, the number of neurons, and the length of the recording? The examples in the paper have latent dimension <= 2, on the order of tens of neurons, and 500 time points (50s binned at 100ms). Is the technique feasible for larger datasets? - The authors ran multiple simulations and show the mean across repeats--the authors should also show the variability (standard error or deviation) across repeats, both for the r^2 performance (error bars) as well as estimated trajectories (error bands). - If I understand correctly, the estimated latent trajectory in both the simulation and hippocampal data tracked the animal's position? That seems surprising--what if you set the latent dimension to 3? or 4, or 1? Minor comments ----------------- - line 103: remove the 'as' in "our model is as a doubly nonlinear ..."